## 'Stressed, uncomfortable, vulnerable, neglected': a qualitative study of the psychological and social impact of the COVID-19 pandemic on UK frontline keyworkers

Tom May 🔘 , Henry Aughterson 🔘 , Daisy Fancourt 🔘 , Alexandra Burton

Research Department of Behavioural Science and Health, Institute of Epidemiology and Health Care, University College London, London, UK

**Correspondence to**
Dr Tom May; t.may@ucl.ac.uk

## ABSTRACT

**Objectives** Non-healthcare keyworkers face distinct occupational vulnerabilities that have received little consideration within broader debates about 'essential' work and psychological distress during the COVID-19 pandemic. The aim of this study was therefore to explore the impact of the pandemic on the working lives and mental health and well-being of non-healthcare keyworkers in the UK.

**Design** In-depth, qualitative interviews, analysed using a reflexive thematic analysis.

**Setting** Telephone or video call interviews, conducted in the UK between September 2020 and January 2021.

**Participants** 23 participants aged 26–61 (mean age=47.2) years employed in a range of non-healthcare keyworker occupations, including transport, retail, education, postal services, the police and fire services, waste collection, finance and religious services.

**Results** Keyworkers experienced adverse psychological effects during the COVID-19 pandemic, including fears of COVID-19 exposure, contagion and subsequent transmission to others, especially their families. These concerns were often experienced in the context of multiple exposure risks, including insufficient personal protective equipment and a lack of workplace mitigation practices. Keyworkers also described multiple work-related challenges, including increased workload, a lack of public and organisational recognition and feelings of disempowerment.

**Conclusions** In efforts to reduce psychosocial concerns among non-healthcare keyworkers, there is a need for appropriate support during the COVID-19 pandemic and in preparation for other infections (eg, seasonal influenza) in the future. This includes the provision of psychological and workplace measures attending to the intersections of personal vulnerability and work conditions that cause unique risks and challenges among those in frontline keyworker occupations.

## INTRODUCTION

In response to the COVID-19 pandemic, restrictions of varying stringency have been imposed by governments around the world to suppress the virus. In the

### STRENGTHS AND LIMITATIONS OF THIS STUDY

⇒ This is the first known qualitative study to interview a range of non-healthcare keyworkers about their experiences of working during the COVID-19 pandemic.
⇒ Data were obtained through in-depth, qualitative interviews with a strong theoretical underpinning between September 2020 and January 2021, thereby complementing earlier quantitative research in this field.
⇒ Findings can inform the development of psychosocial and occupational support for non-healthcare keyworkers, both as COVID-19 persists and in future scenarios.
⇒ Study may be limited by a sample biased towards those motivated or willing to participate.
⇒ Data cover a range of keyworker occupations which, while useful in terms of coverage, may limit specificity.

UK, mitigation measures including self-isolation, mobility constraints and the closure of all but essential workplaces have been implemented in efforts to minimise contact and transmission.[1] While some occupational groups have navigated these measures through flexible working practices (eg, home working) and economic support (eg, 'furlough'), those employed in 'essential' keyworker occupations, including healthcare, transport and education among others, were mostly exempt from such strategies.[2] Consequently, many frontline keyworkers have continued to work throughout the pandemic, often at increased risk of exposure to and acquisition of COVID-19.[3–5]

The psychological demands of working through the COVID-19 pandemic have attracted a substantial amount of academic interest. However, to date, research has

primarily focused on the experiences of health and social care workers, including 'frontline' staff such as nurses, general practitioners (GPs), anaesthetists and care home and social workers.[6–12] These studies have documented elevated levels of stress,[11] anxiety[10] and depression[9] through increased workloads, changing work conditions and feelings of helplessness.[6–12] Health and social care workers have also endured longer working hours with inadequate personal protective equipment (PPE)[7] and have reported fears of infection for themselves and their families.[8 12] There is evidence that previous epidemics (eg, severe acute respiratory syndrome (SARS) and Middle East respiratory syndrome (MERS)) posed similar work-related stressors and subsequent demands on the psychological well-being of those working in health and social care occupations.[13–15] Conversely, there is some evidence that health and social care workers may also experience positive outcomes from working throughout pandemics, including a renewed sense of purpose, contribution and reward.[8 12]

Research investigating the experiences of non-health keyworkers (hereafter 'keyworkers') such as those employed in transport, retail, education and various other public services is limited.[4] Nevertheless, emerging quantitative data suggest that essential service workers (eg, food chain, public security and transport) are experiencing elevated stress and anxiety during the pandemic.[16] A recent publication on grocery store workers in the USA found increased anxiety and depression among employees with direct exposure to customers (eg, cashiers).[4] Correspondingly, a case study of a single UK supermarket employee described how customer behaviours, inadequate PPE and the absence of workplace mitigation policies induced fears of COVID-19 transmission.[5]

Many keyworkers face distinct occupational vulnerabilities that have received little consideration within broader debates about essential work and psychological distress during the pandemic. First, there is evidence that some keyworkers (eg, transport workers) have increased vulnerability to COVID-19 due to older age, the presence of pre-existing health conditions, belonging to a black, Asian or minority ethnic group and residing in an area characterised by high levels of socioeconomic deprivation.[3] Being at increased risk of COVID-19 susceptibility is likely to have a detrimental impact on mental health and well-being due to the perceived negative consequences of infection, as documented in studies with older adults[17] and those with long-term health conditions.[18] Second, many keyworkers, particularly those from low-income, service or elementary occupations, may face financial challenges that increase susceptibility to COVID-19.[2] For example, although the Coronavirus Act 2020 extended Statutory Sick Pay (SSP) to all UK employees, the scheme is based on contractual hours. Part-time employees, or those reliant on overtime, may therefore be unwilling to take leave or self-isolate due to substantial reductions in wages.[2 5] Alternatively, some keyworkers may face financial hardship if they choose to or are required to self-isolate, which may induce mental distress.[19 20]

To date, a large proportion of research on keyworker mental health has been conducted with healthcare workers[6–12] or has focused on specific non-healthcare keyworker groups (eg, grocery store workers).[4 5] However, given that keyworkers fulfil a variety of roles whereby their exposure to the public and potential risk of COVID-19 infection differs,[2 16] there is a need for in-depth qualitative data on a broader range of keyworker experiences and how these may vary among occupations. This is crucial to aid our understanding of specific work-related stressors and to inform future psychosocial support for this group as the COVID-19 pandemic persists and in preparation for other infections (eg, seasonal influenza). To these ends, the study aimed to explore qualitatively the impact of the COVID-19 pandemic on the working lives and mental health and well-being of UK frontline keyworkers.

## METHODS

The research employed a qualitative design using semistructured interviews with UK keyworkers. The study formed part of the UCL COVID-19 Social Study,[21] which explores the psychosocial effects of COVID-19 and associated restrictions on adults in the UK. Participants were interviewed between July 2020 and January 2021 about their working experiences throughout the pandemic, including any implications for mental health and well-being.

### Sample and recruitment

Eligibility was based primarily on whether the person was a non-healthcare keyworker (as defined by UK Government criteria),[22] aged over 18 years, working during the pandemic and living in the UK. Participants were purposively recruited to ensure diversity of gender, age and occupation via social media, personal contacts and the UCL COVID-19 Social Study newsletter and website. Participants were provided with both verbal and written information about the purpose of the research, and informed that their involvement was voluntary. All participants signed a consent form to indicate their agreement to participate and provided demographic information.

### Data collection

Interviews were conducted by TM (research fellow in social science), RC (research fellow in public health) and SE (research assistant) via telephone or video call. All interviewers were experienced qualitative health researchers educated to at least postgraduate level. Interviews followed a topic guide that posed questions about the participant's experience(s) of the impact of

- In what ways has your work life been impacted by the COVID-19 pandemic?
- How do you feel about the changes that have been brought about by Covid-19? Have they had any impact on your mental health or wellbeing?
- Have you been doing/ planning anything to help with this?
- Has the pandemic meant that you have any worries for the future?

**Figure 1** Examples of questions in the topic guide.

the pandemic on work, social life and mental health and well-being. Interviews lasted an average of 45 min and were digitally recorded and transcribed verbatim by a professional transcription service. Interview topic guide development was guided by existing theories on behaviour change,[23] social integration and health,[24] and health, stress and coping.[25] Questions and prompts were designed to illicit responses around: (1) changes to work life, (2) changes to social lives, (3) impact of the pandemic on mental health and (4) worries about the future. Specific topic guide questions are listed in figure 1, and the full topic guide is included in the online supplemental material.

Participants were offered compensation in the form of a £10 high street e-voucher. Data collection continued up until the point at which instances of data emerged consistently, or where no further data would develop new properties, categories or findings (ie, theoretical saturation).[26]

### Patient and public involvement

Participants or members of the public were not involved in the design, conduct or reporting of the study, nor the dissemination of findings. Participants will be provided with study result on request, however. The findings will also be disseminated to the public through social media and newsletters (eg, March Network).

### Data analysis

Following anonymisation by the lead researcher (TM), transcripts were uploaded to NVivo V.12 software for analysis. A reflexive thematic approach was adopted in line with the principles of Braun and Clarke,[27 28] which began with researchers familiarising themselves with data by reading through the individual transcripts. Following this, three transcripts were initially read independently by two researchers (TM and HA), who coded and discussed any emerging codes of potential significance to the research objective. A preliminary coding framework, informed deductively by concepts within the topic guide, was used to guide this process, although an inductive approach was also used to refine the framework in correspondence with any emerging concepts within the data. This was then applied to the remaining transcripts by TM, who reread transcripts and coded and synthesised text into categories, which were subsequently analysed and grouped into themes.

To ensure that the final extracted themes were not just the personal interpretation of one team member, the qualitative research team met weekly to discuss and iteratively refine any new codes or themes that emerged.

## RESULTS

Twenty-three keyworkers were interviewed. Participants were aged 26–61 years, predominantly male (61%) and white British (70%) (see table 1).

Two primary themes were identified: (1) perceptions of personal vulnerability and (2) work-related challenges. These are shown in figure 2, along with their respective subthemes.

### Perceptions of personal vulnerability
#### Fears of contracting COVID-19
The majority of participants relayed fears of contracting COVID-19 while at work. Some had underlying health conditions that heightened these anxieties:

> I was probably more worried than some are, that I might be more prone to catching it. Because I've got asthma, I've got chronic sinusitis, and I just thought, if this is a respiratory thing, you're buggered. (Supermarket worker 1)

Others were less fearful of the implications for themselves but expressed concerns about becoming a source of transmission. Some lived in households with vulnerable family members, including elderly parents and children with underlying health conditions ('*because of my personal circumstances at home, I had two people in their 70s and an asthmatic child. The stress and worry and fear of me basically bringing that home to them was just crippling me*', supermarket worker 2), while others were more concerned about contracting and transmitting the virus to vulnerable members of the public ('*I also don't want to give it to anyone else. I might see someone who's vulnerable, so I'm conscious that it's not me I've got to worry about, it's everyone else*', police staff). Working in environments that posed significant risks to themselves and others was, therefore, a source of anxiety:

> I was so anxious about going to work with the coronavirus. I was quite paranoid. I used to dread leaving the house every day, going into work. It was really, really hard. (Bus driver 1)

**Table 1** Characteristics of participants

| | |
|---|---|
| Number of participants | 23 |
| Profession | Bank worker (1)<br>Bus driver (6)<br>Bus depot supervisor (1)<br>Delivery driver (1)<br>Education staff (deputy head, primary school teacher and teaching assistant) (3)<br>Firefighter (1)<br>Platform staff (1)<br>Police staff (firearms officer, inspector and sergeant) (3)<br>Postal worker (1)<br>Religious staff (2)<br>Supermarket worker (2)<br>Waste operative (1) |
| Age (mean age/range) | 47.2 (26–61) |
| Gender | Male (14)<br>Female (9) |
| Ethnicity | Bangladeshi (1)<br>Black British Caribbean (1)<br>Indian (1)<br>White British (16)<br>White other (Hungarian, Scottish, further data not provided) (3)<br>Other (British Turkish) (1) |
| Month/year of Interview | July 2020 (3)<br>August 2020 (3)<br>September 2020 (9)<br>October 2020 (1)<br>November 2020 (5)<br>January 2021 (2) |

### Exposure risks

Participants noted specific exposure risks at work that prompted fears of contracting COVID-19. Some reported governmental and organisational delays in initiating and implementing protective actions, including workplace instructions aimed at mitigating transmission. As a result, many continued to work without organisational guidance during the initial stages of the pandemic, which prompted feelings of vulnerability:

> So, that first week was really important to me, because we weren't really protected. We didn't know what the crack was about face masks….we were driving around in buses for that week that didn't have protection, what we call an assault screen, you know, something that separates you from the passengers on the bus…and we were thinking, jeez guys, anything could be going on here. (Bus driver 2)

Similarly, most participants reported the inadequate provision of workplace PPE. Some noted initial delays in receiving equipment through their employer ('*hand sanitiser came in, I think, probably three, four weeks after lockdown started*', bus driver 3), while others described limited ('*sometimes we don't even have soap in the bathrooms*', delivery driver) or no supplies ('*we weren't given any kind of PPE. Nothing was offered*', supermarket worker 2). In some workplaces, such as on buses and in supermarkets, other protective measures including daily antiviral cleaning and enhanced sanitation were often inadequate:

> There are aspects of it that worry me. I don't think in some ways [the supermarket] is the most hygienic place in the world. (Supermarket worker 1)

Working closely with the public was an additional concern among some keyworkers. Some noted how some members of the public did not always conform with social distancing guidelines or the wearing of PPE ('*there are people not getting on with masks when they should, or if they are wearing one they are wearing one under their chin. I would say 80% of people are being compliant, but then you've got 20% of people who don't give a monkeys*',

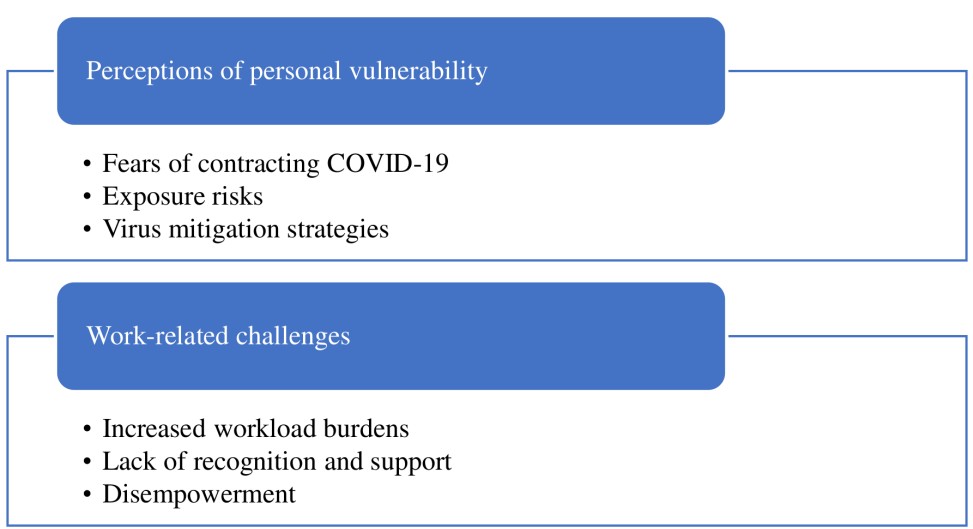

**Figure 2** Key themes.

bus driver 4). Others reported how the public would also, at times, behave inappropriately around staff. This was often frightening for participants:

> I mean, we did initially have some young lads come in who were actually deliberately coughing and sneezing, both on colleagues and other customers. And it really freaked a lot of people out because people were genuinely in fear. (Supermarket worker 2)

Relatedly, some keyworkers worked in confined spaces that were unconducive to social distancing ('*social distance is quite hard at the depot to do*', delivery driver), or worked with colleagues who did not follow social distancing rules. The inability to properly socially distance elevated fears of potential exposure:

> I don't feel very safe… because many, many drivers arrive and they meet with other people as well and I don't know where they are or who they are …a few of them was coughing…and they said, oh it's just a cold. But you think it's a cold but how I supposed to know that it's not. (Delivery driver)

### Virus mitigation strategies

To mitigate concerns about contracting and transmitting the virus, participants often enacted their own mitigation strategies. Some reported purchasing and wearing their own PPE ('*I got my face mask, I got a cloth one… I have started wearing a hoodie as well, just to cover me whole*', bus driver 6) and sanitising their workspace ('*I took my own bleach solution and soapy water solution and was cleaning everything in the cab… we were all bringing our own stuff in… just to be safe*', bus driver 4). Such measures were acted out in the absence of inadequate PPE provision:

> There was no hand sanitisers. There was nothing. Absolutely zero. Even during lockdown, for the first part of it, there was nothing at all. It was down to the drivers. (Bus driver 4)

While these measures enabled participants to psychologically cope with stressful working conditions, they did not always prevent family members or loved ones from feeling anxious about possible transmission. To reduce these concerns, some keyworkers would therefore 'decontaminate' on re-entering their home:

> So when I come from school, I literally strip off at the door. Everything goes into a bag, everything gets cleaned off. I don't talk to anyone or touch anyone. I don't go near anyone until I've decontaminated (teacher 1).

Others temporarily separated from anxious loved ones by either sleeping in separate bedrooms ('*[husband] went in the spare room, so he kind of lived in the spare room for a long time, so that we were distanced*',

supermarket worker 1) or moving out of their home. One bus driver, for example, moved to rented accommodation to protect his wife from the risk of infection. Such measures, while deemed necessary by participants, induced additional psychosocial strains including loneliness and isolation:

> [I feel] Very lonely…I've been with [wife] since 1990. We've always been together, always done things together and to suddenly be sitting in a room on your own is quite dire. It upset me at first. I cried myself to sleep for a few nights, you can't believe this is happening. (Bus driver 4)

### Work-related challenges
#### increased workload burdens

The pandemic presented several work-related disruptions and challenges. Staff who were infected with COVID-19 or had been in close contact with a case were required to self-isolate. This often resulted in staff shortages:

> During lockdown, we were decimated with staff. We were absolutely on our backside… so, I was actually out on weekends, on Saturdays, driving vehicles supporting the operation leaders. We didn't have enough staff. (Waste operative)

Insufficient staff numbers resulted in increased workloads and longer hours, often without extra pay ('*we're doing more hours. They increased the length of the shift. We're on a salary. We're not hourly-paid so obviously, when we were due to do a shorter shift we would still get a long one*', bus driver 4). Some participants were also required to perform additional or new duties to relieve workload burdens, which were an additional source of stress:

> We're totally doing jobs that we never did before, because we're answering the telephone calls, because our call centre is in India, and they're on total lockdown… so, that part I find stressful. (Bank worker)

The stress of increased workload burdens and carrying out new tasks beyond usual levels of expertise would, at times, lead to tension and conflict within the workplace:

> A lot of friction, people just snapping at each other over the slightest thing. It would just set people off. A couple of times, I had to intervene. Guys, calm down. Jesus, boys. What are you doing?… behave yourself… I was having to stop people pulling lumps out of each other. (Waste operative)

Additionally, those who transitioned to online working (including police, teachers and bank workers) welcomed such changes but noted difficulties. Tasks that were previously performed with ease proved more challenging when working from home (eg, communicating with colleagues). Some also reported being 'overloaded' with virtual meetings:

> Because it's virtual and I chaired a meeting the other day and I said, look, I need to eat, I need to get up.

Because what you don't see is, we have a meeting here now, and then say yes, bye, and then I'm straight into another one…so I think there's been a huge overload. (Police staff 2)

### Lack of recognition and support

Although some participants were appreciative of the support they received from the public, some felt undervalued, particularly in comparison with National Health Service healthcare workers whose work was recognised regularly in public gestures of appreciation (eg, clap for carers):

They deserve the respect they get, the NHS people, and they should. But I think a lot of people forgot about there's people out there like myself on the railway, bus drivers as well. And there's been really, not much for people, like myself, in the frontline. (Platform staff)

Internal recognition (ie, from management) was also limited ('*Internally, from management… I don't think the recognition has been as wide as it could or should be*', waste operative). In particular, keyworkers felt that the risks they were exposed to were not fully acknowledged or appreciated ('*I felt stressed. I felt uncomfortable. I felt vulnerable. I felt neglected. I felt everything because the company still don't think it's serious*', bus driver 5). Some felt that profit was sometimes prioritised over staff safety:

Management don't give a crap about staff. They just care about the things that goes in the till, which is the money. And they don't want to pay sick pay. There was another one… his wife was a teacher and she was told to self-isolate. So obviously, he should have been self-isolating, because there was an outbreak at the school. He was told by the manager of the store just to come in, it wasn't a problem. (Supermarket worker 2)

### Disempowerment

Despite concerns about contracting COVID-19, many participants felt that they had to work for fear of financial implications or punitive measures. Some were concerned that protracted absences would result in disciplinary action ('*But the particular academy chain that I work for has said that if teachers are not available to work from day one when they come back, then it will be disciplinary*', teacher 1) or job loss:

People were genuinely scared because the government was saying this and your manager's going, no, you do this or you don't have a job…you can't afford not to be there or to lose hours or to lose your job. (Supermarket worker 2)

Participants reported opportunities to take furlough or sick leave but noted the financial implications of doing so. For example, some participants (particularly supermarket workers, bus drivers and police staff) relied on overtime

to supplement their income. However, additional hours are not accounted for in SSP or furlough schemes. Any absence would subsequently result in financial hardship:

I worked all the way through since the beginning. I was given the choice of furlough, but I turned it down…it would have been such a drop in money, it would have put a financial hardship on us. (Bus depot supervisor)

In this context, many keyworkers recognised that they had no option but to continue working ('*I just thought, well, I either stay at home and do nothing and go unemployed, or I carry on working. And that was literally my two options. There was no middle*', bus driver 3). Some reported feeling powerless and resigned themselves to the possibility of contracting COVID-19:

And in my line of work, being on the frontline, there's probably a high chance that I am going to probably get it at some point. And you just resign yourself to the fact. (Platform staff)

## DISCUSSION

The findings presented in this paper are particularly valuable as, to date, non-healthcare keyworker voices are largely absent within broader debates about 'essential' work and psychological distress during the COVID-19 pandemic.[4] Therefore, this study provides new insights into the psychological impact of frontline work during the COVID-19 pandemic, including how non-healthcare keyworkers respond to and experience previously identified occupational risks, including insufficient PPE[2] and the inability to socially distance.[4]

By far the most prevalent stressor was the fear of contracting COVID-19. Those who continued to work close to others or in environments unconducive to social distancing reported feelings of exposure and vulnerability. Consistent with research with health and care workers, feeling unsafe and vulnerable to infection are predictive of poor mental health.[9 29] Frontline health and social care workers, for example, were more likely to experience greater psychosocial distress during the COVID-19 pandemic and previous outbreaks because they were likely to have the most direct patient contact.[12 29 30] This is not dissimilar from recent data documenting elevated psychological distress among supermarket workers unable to socially distance at work during the COVID-19 pandemic.[4 5] Although it appears a similar awareness of one's vulnerability increased feelings of anxiety among our sample, our findings highlight additional occupational factors and working conditions that compounded fears of contagion, including the inadequate provision of PPE and organisational delays in initiating and implementing protective actions aimed at mitigating transmission.

In response to these risks, many participants enacted their own risk reduction practices, including purchasing PPE, sanitising their workplaces and temporary separation

from family members. While such measures helped reduce feelings of exposure, they also reinforce widespread concerns from keyworkers and public health officials regarding the inadequacy of PPE provision for those in frontline occupations during the pandemic.[2 31] This is potentially concerning for the well-being of keyworkers, given that previous research has highlighted how precautionary workplace measures, including sufficient PPE and infection control measures, are associated with decreased levels of concern and emotional exhaustion among healthcare workers.[29 32] The provision of protective measures by employers is also likely to reduce the need to enact mitigation strategies (eg, temporary separation) that may trigger additional psychosocial burdens (eg, loneliness and isolation).[10]

Workplace challenges also posed several additional stressors. Increased workloads were common and led to elevated feelings of stress and subsequent workplace tension and conflict. Some participants also reported limited internal recognition for their work and felt that the risks they were exposed to were not fully acknowledged by senior staff. Although workplace unity has been found to be an important source of support and resilience among health and social care workers during the COVID-19[10 12] and previous pandemics,[15 33] this protective factor was therefore not experienced by keyworkers in our study. Similarly, while health and social care workers may experience comparable workload challenges, these are often endured alongside enhanced public and organisational recognition for their efforts (eg, clap for carers). Among health and social care workers, greater recognition—both publicly and organisationally—has been shown to produce protective mechanisms linked to resilience, including a renewed sense of purpose, contribution and reward.[8 12] The absence of similar public and organisational appreciation limited the emergence of any 'positive' psychosocial effects occurring among those in our study. Hence, many keyworkers experienced workplace challenges in the absence of protective and support mechanisms proven beneficial to other occupational groups.[16]

Many participants reported feeling powerless to the situation. This was primarily due to fears of financial hardship or disciplinary action. Indeed, there is evidence that some keyworkers, particularly those part time or heavily reliant on overtime, may be unwilling to take leave or self-isolate due to substantial reductions in wages.[5] Many participants reported similar concerns and that they had no option but to continue working, despite concerns about possible infection. Conversely, those who did take leave, whether through SSP or furlough, reported income losses. This is a particular concern given how COVID-19 induced economic hardship is having adverse effects on the psychological well-being of the population.[34–37]

These findings should be considered in light of a number of limitations. First, while this study provides unique and important insights into keyworkers' experiences during the pandemic, the timing of the interviews may need to be considered when interpreting the findings. The majority of interviews were conducted between September and November 2020. While this meant that participants were able to recount both current and retrospective experiences during periods of lockdown and more relaxed measures, as the pandemic is ongoing, experiences are still evolving. Second, this study may be limited by a sample biased towards those motivated and willing to participate. There is the potential that the views and experiences of those unable or unwilling to participate may differ from those in this study (eg, unaffected by working conditions) and have therefore not been documented. Finally, our data cover a range of keyworker occupations which, while useful in terms of coverage, may limit specificity. Where possible, we have attempted to draw out any distinctions between occupations in the reporting of our results.

Our study has some important implications for policy and organisational practices. First, our findings suggest that sufficient protective measures in workplaces are urgently required, as many participants reported feeling exposed and unsafe. The inadequacy of governmental and organisational responses to the pandemic is highlighted by the fact that some enacted their own mitigation practices to prevent exposure to and acquisition of COVID-19. Hence, the provision of adequate PPE, strategies aimed at reducing interpersonal contact (including temporary accommodation, as has been provided for some healthcare workers),[38] and repeat and routine employee testing are but a number of measures that should be pursued to safeguard keyworkers who continue to operate on the frontline.[2] For keyworkers who are most at risk, an increased range of actions is needed to protect them from exposure, given that the most vulnerable workers (whether due to underlying condition, age, ethnicity or financial situation) reported the greatest concerns regarding work-related stressors. Second, adequate and accessible financial support must be provided to safeguard keyworkers' health during this pandemic and beyond. This is especially important for those keyworkers who, due to the nature of their job, are unable to access furlough schemes or sick pay because of worries about financial loss.[39 40] Third, learning from the experiences of keyworkers in other occupations (eg, health and social care workers) may assist with planning interventions designed to assist resilience in pandemics. Some health and social care workers have noted the importance of public recognition and social support in minimising the psychological impact of the COVID-19 pandemic[12] and other infectious disease outbreaks.[29] Our data suggest a need to provide similar recognition for those working in occupations detailed in this study to buffer negative psychological consequences. Finally, while these measures may help mitigate the immediate psychological effects of the pandemic, it is worth noting that previous research conducted before the pandemic has identified similar psychological demands among keyworkers to those highlighted in this article, including occupational stress,[41–43] low levels of job satisfaction[42 44] and burnout.[45] Hence,

although support for keyworkers is needed now more than ever, workplace support packages must be provided beyond this period to address long-standing problems for those employed in keyworker occupations.

## CONCLUSION

This study highlights the psychological impact of the COVID-19 pandemic on those employed in frontline keyworker occupations in the UK. Participants reported anxiety about COVID-19 exposure and transmission to others, especially their families. These fears were often endured in the context of multiple exposure risks, including insufficient PPE and workplace support. Keyworkers also experienced work-related challenges, including increased workloads, a lack of recognition and a sense of helplessness. This study therefore contributes to understandings of how the intersections of personal vulnerability and work conditions produce unique risks and challenges among those in frontline occupations.

It is hoped that by recognising the voices of those who do not feel adequately supported, protected or valued for their work may be an initial step in understanding the psychosocial and occupational support non-healthcare keyworkers need, both as COVID-19 persists and in similar future scenarios.

**Acknowledgements** The authors would like to thank Dr Rana Conway and Sara Esser for their help with interviewing participants as part of this study, and Joanna Dawes, Dr Alison McKinlay, Dr Anna Roberts and Dr Katey Warran for their assistance during the recruitment and analysis stages of this paper. The authors would also like to thank those people who gave up their time to take part and contribute to the study.

**Contributors** DF conceived the initial study, and DF and AB contributed to the study design and ethical approval process. TM was responsible for data collection and was assisted by RC and SE during this stage. TM conducted formal analysis alongside HA, who coded three transcripts for cross-checking purposes. TM produced the original draft of the manuscript, which HA, AB and DF critically reviewed and edited. All authors approved the final manuscript for submission. TM is responsible for the overall content as guarantor.

**Funding** This COVID-19 Social Study was funded by the Nuffield Foundation (WEL/FR-000022583), but the views expressed here are those of the authors. The study was also supported by the MARCH Mental Health Network funded by the Cross-Disciplinary Mental Health Network Plus initiative supported by UK Research and Innovation (ES/S002588/1) and by the Wellcome Trust (221400/Z/20/Z). DF was funded by the Wellcome Trust (205407/Z/16/Z).

**Competing interests** None declared.

**Patient consent for publication** Not applicable.

**Ethics approval** The study was reviewed and approved by the University College London Ethics Committee (Project ID 14895/005).

**Provenance and peer review** Not commissioned; externally peer reviewed.

**Data availability statement** No data are available. The data are not publicly available due to their containing information that could compromise the privacy of research participants.

**ORCID iDs**
Tom May http://orcid.org/0000-0003-3077-523X
Henry Aughterson http://orcid.org/0000-0001-5568-6474
Daisy Fancourt http://orcid.org/0000-0002-6952-334X

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
