## [Reviewer comments · BMJ Open]

ARTICLE DETAILS

TITLE (PROVISIONAL)	'Stressed, uncomfortable, vulnerable, neglected': a qualitative study of the psychological and social impact of the COVID-19 pandemic on UK frontline keyworkers.
AUTHORS	May, Tom; Aughterson, Henry; Fancourt, Daisy; Burton, Alexandra

VERSION 1 – REVIEW

REVIEWER	Reilly, Shannon E University of Virginia Health System, Neurology
REVIEW RETURNED	07-May-2021

GENERAL COMMENTS	Thank you for the opportunity to review this manuscript, 'Stressed, uncomfortable, vulnerable, neglected': a qualitative study of the psychological and social impact of the COVID-19 pandemic on UK frontline keyworkers." It was a pleasure to read this work, which is important and timely. The authors set up the aims well in the introduction, describe their findings in a lovely narrative, and draw logical implications and conclusions from the data. There are some areas for improvement in the manuscript (particularly in the reporting of the Methods) that I believe the authors could address in a minor revision. Specifically, the authors should consider describing their hypotheses at the end of the introduction, and more in-depth descriptions of the interview guide and theoretical underpinnings are warranted. My comments regarding each section are detailed below. ABSTRACT Overall, the abstract is clear, concise, and well written but would benefit from a few tweaks. Similar to in the introduction, it would be helpful to briefly describe the authors' hypothesis. In the Results section, I don't think "psychological effects" is the best way to term the first theme for a couple of reasons: that is not how it is described in the paper, and in the paper the second theme (workplace-related challenges) has many psychological components (e.g., feelings of disempowerment, feeling undervalued). INTRODUCTION Overall, the introduction has a good flow and sets the stage nicely for the aims of the study. However, one of the strengths of the study described in the "strengths and limitations" section is that this study builds on prior quantitative research and that there has not been any qualitative study like this. It would strengthen the introduction and the novelty of the study to flesh this out a bit more in the introduction. The end of the introduction is also missing hypotheses. What themes did the authors expect to find in these qualitative interviews, and to what extent did the findings match hypotheses (in the introduction)? Even if the study was exploratory, it is likely that the authors had general expectations as to what they might find.
--

On p. 5, line 34, has the research on HCWs been quantitative, qualitative, or both? If only quantitative, it may be good to include that to further differentiate prior work from this study.

“Health care workers” is a readily understandable term that does not need explanation, but “health and social care workers” (p. 5, line 36) would benefit from an additional clause describing what types of workers are included in this phrase. And, if the phrase is health and social care workers, perhaps the acronym should be HSCW instead of HCWs.

I appreciate the inclusion of evidence from prior pandemics, as well as evidence about possible “silver linings” associated with working through a pandemic.

In the first paragraph of p. 6, the two examples of non-healthcare essential workers both involve grocery stores. Are there any other prior studies that could be added to provide coverage of a wider variety of workplace settings?

On p. 6, line 24, it is unclear what is meant by “Despite these similarities” – to healthcare essential workers? Please clarify what you mean by this.

On p.6, lines 28-33 when discussing increased vulnerabilities that some essential workers have, there needs to be more clarification here. Pre-existing health conditions is self-explanatory, but the other three characteristics need to be more specific – older age? Which areas of residence? Which ethnicities and why might this be?

On p. 6, lines 42-45, how do financial challenges increase susceptibility to COVID-19? Is it related to the following couple of sentences about having more exposure due to inability to reduce work hours (which is a good point)? If so, please make this connection clearer, and if you were getting at something different in that first sentence, it would be helpful to expand on this a bit more.

METHODS

Upon quick review, it seems that the UCL COVID-19 Social Study is a much larger study (N = ~45,000). If this is the case, it would be helpful to state that, particularly given the small sample size for these in-depth interviews. How was this particular sample chosen out of the larger sample for this particular study? In the COREQ Checklist, the authors state “N/A” for who refused to participate/dropped out. Did all recruited participants agree to participate in the study? If so, it would be good to state this explicitly in the Methods and if not, the authors should detail this in the Methods.

*There is not enough information about the interview guide/topic guide on p. 9 (e.g., how it was developed, who was involved in developing it, how it was based upon prior research, whether it was piloted). It would be helpful to have examples of some of the questions asked in the methods section so that the reader has a better understanding of what to expect when going into the Findings.

It would be helpful to have a bit more information about where the interviews were conducted. Of course it was via phone or video, but were participants typically in their homes or somewhere else? What

percentage of interviews were via phone versus video, and were there any differences detected (e.g., in interviewer-participant rapport, quantity or quality of information provided) between the two modalities?

*I am not a qualitative researcher, so I cannot comment from an informed position on the strength of the data analysis section. However, the abstract mentions that the authors used reflexive thematic analysis as the design, and this is not mentioned in the Methods section. It would be helpful to provide more background about the theoretical underpinnings of this and what it looks like procedurally, particularly because the authors note in the strengths section that the interviews had “a strong theoretical underpinning.” There needs to be more evidence of this in the Methods section.

Additionally, it would be a good idea to briefly describe the data privacy and security entailed in uploading participants’ transcripts to NVivo. Upon quick review, it seems like this service encrypts files and stores data securely.

FINDINGS/RESULTS

In the Abstract, this section is termed “Results” – it would be good to keep the terminology consistent throughout the paper.

Table 1 should be referenced in-text at the beginning of the Findings section. In Table 1, it is unclear whether 47.2 is the mean age of participants. Please clarify this, and it would be helpful to have the standard deviation as well. Regarding descriptions of Ethnicity, it seems exclusive to use the term “Other,” so the authors may consider a different phrasing. It is unclear what the distinction is between “White British” and “White Other.” Does the “Other” in “Black Other” and “White Other” mean “non-British”? If so, it would be helpful to phrase it that way. For the final category, rather than saying “Other” it may be more appropriate to say “A different ethnic identity.”

On p. 12, line 39, it would be helpful to quantify what is meant by “the majority” (e.g., what percentage of participants)? Same comment for p. 14, line 15: “most participants reported the inadequate provision of workplace PPE.”

It’s unclear to me why some of the direct quotes are integrated into the paragraphs and some are separated out. It does not seem to be related to the length of the quote.

I appreciate the narrative that the authors crafted in describing the qualitative results – each section leads well into the next.

On p. 16, line 52, the authors write “whilst deemed necessary” about the measures of participants separating from loved ones. The authors should clarify that the participants deemed this necessary, if that was the case.

DISCUSSION

The authors did a wonderful job writing this discussion overall.

On p. 22, line 27, the following claim needs a citation: “Consistent with research with HCWs, feeling unsafe and vulnerable to infection are predictive of poor mental health.”

	On p. 23, line 15, I just wanted to clarify whether the authors meant “provision of protective measures by employees” or “by employers.” It seems that “employers” makes more sense if institutional provision of these measures would decrease stress and isolation. But perhaps the authors mean that when the employers are not providing this and the employees themselves provide them (e.g. hand sanitizer, etc.), it has the same outcome on mental health. On p. 23, line 28, I appreciate that the authors brought up that a theme of workplace unity was not found in this study. It would be helpful to expound on this a little as to the reasons why that might be. For instance, it seems like many of the jobs represented in the study were more solitary positions (e.g., bus drivers) that may not engender as much of a sense of community. On p. 24, line 23, a stronger statement could be made – the timing of the study should be considered when interpreting the findings, particularly because of the rapidly evolving/changing nature of COVID-19 over the past year. The authors note at this point in the discussion that the majority of interviews were completed between September and November 2020. It would be helpful to add to Table 1 a frequency of which month the interviews took place in from September 2020 to January 2021 to have a better sense of the context of the majority of these interviews. On p. 24, line 34, the participants were motivated “and” willing (rather than “or”), given that they did in fact participate. On line 39, data is plural, so the verb should be “cover” instead of “covers.” The authors make the comment on p. 24, line 43 that “Where possible, we have attempted to draw out any distinctions between occupations in the data.” I am not sure what they are referring to exactly, and it would be helpful to discuss any distinctions (whether in the Results or Discussion), particularly because a large percentage of the sample was bus drivers, compared to other occupations that generally had only 1 participant. On p. 25, line 42, the authors state that “it is worth noting that many of the psychological demands experienced by keyworkers and highlighted in this article existed well before the advent of COVID-19, including occupational stress³⁴⁻³⁶, low levels of job satisfaction^{35 37} and burnout³⁸.” While this is an interesting idea to bring up and could be relevant for future implications, I am not sure whether this is supported by the findings. It would be helpful to add an example or two here as to how the data support this conclusion.
--	---

REVIEWER	Ghosh, Shilpi Visva-Bharati University, EDUCATION
REVIEW RETURNED	07-May-2021

GENERAL COMMENTS	REVIEW REPORT ID of the article: bmjopen-2021-050945 Title of the Article: ‘Stressed, uncomfortable, vulnerable, neglected’: a qualitative study of the psychological and social impact of the COVID-19 pandemic on UK frontline workers. Authors: Tom May*, Henry Aughterson, Daisy Fancourt, Alexandra Burton Report: Observations: 1. The topic chosen for the investigation is utmost significant in the
--

	present day scenario. The design of the investigation is indeed one of the pioneer efforts.  2. The rationale of the study is well-written. 3. The procedure for sample recruitment was done appropriately. Ethical criteria were addressed. 4. The process of theoretical saturation was accurately followed for data collection. 5. Empirical process was followed for data analysis. All the themes and sub-themes have been analyzed meticulously. 6. The strengths and limitations of the study have been explained with clarity. 7. All the sub-themes have been adequately addressed in the discussion. 8. References are appropriate and well researched. Suggestions:  1. In the abstract on page no. 03 line no. 28 five professions have been mentioned whereas in the findings on page no. 11 line nos. 17-34 thirteen professions have been mentioned. Therefore in the abstract either the terms 'and others' may be incorporated with the previously mentioned five professions or line no. 25 on page no. 03 may be rephrased and written as '... in a range of 13 non-healthcare keyworker occupations...' 2. Age- range of the sample should be mentioned in the abstract. 3. On Page no. 15 line nos. 50-55 and Page no.16 line nos. 03-11 the investigators have very well addressed the 'feelings of loneliness and isolation of the participants' very well. However, one or two interviews of the participants depicting that they are missing their children may enrich this finding further. 4. Few more references (03-05) may be incorporated in the discussion. 5. Few more studies from Asian, African and South American countries may be referred (if possible) for the present empirical study and incorporated in the references.
--	--

REVIEWER	Williams, Julia University of Hertfordshire
REVIEW RETURNED	10-Aug-2021

GENERAL COMMENTS	This is a unique, interesting and valuable piece of work. As you identify the emphasis in research has largely been on HCWs' experiences during COVID19 rather than the impact on non-health care key workers. There are some fascinating parallels with literature emerging about HCWs' experiences during COVID19.and, equally, there are clearly some interesting differences. The narrative around the social construction of responses to personal vulnerability during this pandemic is truly informative and will benefit from deeper exploration. I agree with your comments about having included such a diverse set of jobs/professions that specificity may be limited but, what you have done is highlight some of the shared experiences and feelings that your participants have described. It would have been interesting to have included your semi structured interview guide so that readers can see what the specific focus was during your interviews. Absolutely worthy of being in the public domain - a strong piece of exploratory, qualitative research which increases our understanding of non healthcare workers' experiences of being on the 'frontline' during the COVID19 pandemic.
---

	One tiny point is that , at times, you refer to data in the singular - as I say a tiny point to consider. Thank you.
--	--

VERSION 1 – AUTHOR RESPONSE

Reviewer: 1	
Overall, the abstract is clear, concise, and well written but would benefit from a few tweaks. Similar to in the introduction, it would be helpful to briefly describe the authors' hypothesis. In the Results section, I don't think "psychological effects" is the best way to term the first theme for a couple of reasons: that is not how it is described in the paper, and in the paper the second theme (workplace-related challenges) has many psychological components (e.g., feelings of disempowerment, feeling undervalued).	Thank you for reviewing the paper and your kind comments. We have amended the paper based on your suggestions, which have helped significantly strengthen the paper. Please see below amendments/comments to your suggestions below.
Introduction	
Overall, the introduction has a good flow and sets the stage nicely for the aims of the study. However, one of the strengths of the study described in the "strengths and limitations" section is that this study builds on prior quantitative research and that there has not been any qualitative study like this. It would strengthen the introduction and the novelty of the study to flesh this out a bit more in the introduction. The end of the introduction is also missing hypotheses. What themes did the authors expect to find in these qualitative interviews, and to what extent did the findings match hypotheses (in the introduction)? Even if the study was exploratory, it is likely that the authors had general expectations as to what they might find.	Thank you for the suggestion. We have added further quantitative work (Paul et al., 2021) (see page5/6) in the introduction to further highlight the dominance of quant. work in this field. We have also fleshed out the end of the introduction with more about the need for qualitative work that builds upon this previous work. "To date, a large proportion of research on keyworker mental health has been conducted with healthcare workers⁶⁻¹² or has focused on specific non-healthcare keyworker groups (e.g. grocery store workers^{4,5}). Given that keyworkers fulfil a variety of roles whereby their exposure to the public and potential risk of COVID-19 infection differs^{2,16}, there is a need for in-depth qualitative data on a broader range of keyworker experiences and how these may vary among occupations. This is crucial to aid our..." In qualitative research, it would be unusual to present hypotheses or objectives, rather, qualitative inquiry usually begins with a "central question" with the intention being to explore complexity surrounding a "central" phenomenon (in our study - working lives, mental health and wellbeing during the COVID-19 pandemic) through the perspectives of certain individuals or groups (in our case key workers). Our central research question is therefore presented at the end of the introduction section on page 6
On p. 5, line 34, has the research on HCWs been quantitative, qualitative, or both? If only quantitative, it may be good to include that to further differentiate prior work from this study.	Thank you for highlighting this. We have drawn out where we are referring to either quantitative or qualitative work throughout the introduction to differentiate our study.

“Health care workers” is a readily understandable term that does not need explanation, but “health and social care workers” (p. 5, line 36) would benefit from an additional clause describing what types of workers are included in this phrase. And, if the phrase is health and social care workers, perhaps the acronym should be HSCW instead of HCWs.	Thank you for this suggestion. To avoid ambiguity, we have removed the abbreviation ‘HCWS’ and referred to instead either ‘health and social care workers’ or ‘health care workers’ based on the focus of the cited papers: studies have predominantly focused on either a combination of health and social care workers (For example -- Augherston et al, 2021) or healthcare workers Vimercati et al 2020). This is now clarified within our paper.
I appreciate the inclusion of evidence from prior pandemics, as well as evidence about possible “silver linings” associated with working through a pandemic.	Thank you 😊
In the first paragraph of p. 6, the two examples of non-healthcare essential workers both involve grocery stores. Are there any other prior studies that could be added to provide coverage of a wider variety of workplace settings?	We were unable to find any further peer reviewed literature with essential workers beyond those employed in grocery stores. We have, however, added findings from a quantitative study (Paul et al., 2021) that investigates essential service workers collectively (including food chain, public security and transport) to expand coverage here.
On p. 6, line 24, it is unclear what is meant by “Despite these similarities” – to healthcare essential workers? Please clarify what you mean by this.	Thank you for the suggestion. We agree this phrase was confusing and have therefore removed it.
On p.6, lines 28-33 when discussing increased vulnerabilities that some essential workers have, there needs to be more clarification here. Pre-existing health conditions is self-explanatory, but the other three characteristics need to be more specific – older age? Which areas of residence? Which ethnicities and why might this be?	Thank you for the suggestion. We have amended to specify the characteristics that may increase vulnerability to COVID-19. “First, there is evidence that some keyworkers (e.g. transport workers) have increased vulnerability to COVID-19 due to older age, the presence of pre-existing health conditions, belonging to a Black, Asian or Minority ethnic group and residing in an area characterised by high levels of socioeconomic deprivation³”
On p. 6, lines 42-45, how do financial challenges increase susceptibility to COVID-19? Is it related to the following couple of sentences about having more exposure due to inability to reduce work hours (which is a good point)? If so, please make this connection clearer, and if you were getting at something different in that first sentence, it would be helpful to expand on this a bit more.	Thank you for the comment. I have added a ‘for example’ to make the connection between the point (financial challenges increased susceptibility) and example (inability to reduce work hours) clearer. “Second, many keyworkers, particularly those from low-income, service, or elementary occupations, may face financial challenges that increase susceptibility to COVID-19¹¹. For example, although the Coronavirus Act 2020 extended Statutory Sick Pay (SSP) to all UK employees, the scheme is based on contractual hours. Part-time employees, or those reliant on overtime, may therefore be unwilling to take leave or self-isolate due to substantial reductions in wages^{9 11}”
Methods	
Upon quick review, it seems that the UCL COVID-19 Social Study is a much larger study (N = ~45,000). If this is the case, it would be helpful to state that, particularly given the small sample size for these in-	Thank you for looking into this. You are correct in that the UCL COVID-19 Social Study comprises a larger quantitative study. A qualitative arm (upon which this study is based) forms a separate component and participants were not recruited directly from

depth interviews. How was this particular sample chosen out of the larger sample for this particular study? In the COREQ Checklist, the authors state “N/A” for who refused to participate/dropped out. Did all recruited participants agree to participate in the study? If so, it would be good to state this explicitly in the Methods and if not, the authors should detail this in the Methods.	the aforementioned quantitative component, but instead through a broader recruitment strategy via social media, personal contacts and the UCL COVID-19 Social Study newsletter and website (page 7). As we recruited a non-random, purposive sample, we were therefore reliant on participants volunteering to participate. As such, we did not collect data on this information.
*There is not enough information about the interview guide/topic guide on p. 9 (e.g., how it was developed, who was involved in developing it, how it was based upon prior research, whether it was piloted). It would be helpful to have examples of some of the questions asked in the methods section so that the reader has a better understanding of what to expect when going into the Findings.	Thank you for the suggestion. Further information regarding the interview topic guide development is now included in the methodology, and some specific questions are now included in Figure 1. For the full topic guide, this is included in supplementary material. Interview topic guide development was guided by existing theories on behaviour change²³, social integration and health²⁴, and health, stress and coping²⁵. Questions and prompts were designed to illicit responses around (i) changes to work life, ii) changes to social lives, iii) impact of the pandemic on mental health and iv) worries about the future. Specific topic guide questions are listed in Figure 1, and the full topic guide is included in the supplementary material.
It would be helpful to have a bit more information about where the interviews were conducted. Of course it was via phone or video, but were participants typically in their homes or somewhere else? What percentage of interviews were via phone versus video, and were there any differences detected (e.g., in interviewer-participant rapport, quantity or quality of information provided) between the two modalities?	Thank you for highlighting this. We did not record where interviews were conducted (e.g. home) as we did not feel it pertinent to the study. No noticeable differences were detected across the interviews in terms of rapport, quantity or quality of interview.
*I am not a qualitative researcher, so I cannot comment from an informed position on the strength of the data analysis section. However, the abstract mentions that the authors used reflexive thematic analysis as the design, and this is not mentioned in the Methods section. It would be helpful to provide more background about the theoretical underpinnings of this and what it looks like procedurally, particularly because the authors note in the strengths section that the interviews had “a strong theoretical underpinning.” There needs to be more evidence of this in the Methods section.	Thank you for the comment. To clarify, we have added in a sentence to highlight our approach (reflexive thematic) prior to the procedures taken in the analysis. The procedures are listed after this sentence: “A reflexive thematic approach was adopted in line with the principles of Braun and Clarke, which began...” Reference has also been made to two papers by Braun and Clarke, who recommend the procedures for analysis listed.
Additionally, it would be a good idea to briefly describe the data privacy and security entailed in uploading participants’ transcripts to NVivo. Upon quick review, it seems like this service encrypts files and	All transcripts were anonymised and de-identified prior to being uploaded to anNvivo database for analysis, as stated on page 8. As Nvivo is a database held on secure UCL servers rather than a service requiring data to be encrypted and stored securely, it would not be necessary

stores data securely.	to describe the data privacy and security entailed in uploading transcripts.
Findings	
In the Abstract, this section is termed “Results” – it would be good to keep the terminology consistent throughout the paper.	Thank you for spotting this. I have amended to ‘results’ at the start of the section.
Table 1 should be referenced in-text at the beginning of the Findings section. In Table 1, it is unclear whether 47.2 is the mean age of participants. Please clarify this, and it would be helpful to have the standard deviation as well. Regarding descriptions of Ethnicity, it seems exclusive to use the term “Other,” so the authors may consider a different phrasing. It is unclear what the distinction is between “White British” and “White Other.” Does the “Other” in “Black Other” and “White Other” mean “non-British”? If so, it would be helpful to phrase it that way. For the final category, rather than saying “Other” it may be more appropriate to say “A different ethnic identity.”	Thank you for the comment. Reference has been made to the table at the start of the findings. We have also clarified that this is the mean age and range. We appreciate the complexities associated with terminology pertaining to ethnic and racial minorities. We have chosen to list descriptions as how participants self-identified. For clarity, however, we have defined what is meant by ‘other’ (e.g. Hungarian, Scottish and for one participant no further details were provided therefore we have listed them as “Further data not provided”).
On p. 12, line 39, it would be helpful to quantify what is meant by “the majority” (e.g., what percentage of participants)? Same comment for p. 14, line 15: “most participants reported the inadequate provision of workplace PPE.”	Although we appreciate that conveying quantitative information in reports from qualitative work has benefits (e.g., greater transparency, increased precision of statements), it is common for qualitative researchers to instead semi-quantify data through terms such as ‘many’ or ‘most’. We have chosen to do this for a few reasons: (1) not everyone was asked the same question in the same way. Reporting frequency of a given response may therefore misrepresent data and lead to generalizations about findings. (2) quantifying responses may detract from the nuanced way responses may vary between participants, and (3) presenting findings as numbers can imply a measurable and objective approach that conflicts with the aims of qualitative research. The type of semi-quantification in our study (e.g. ‘some’, ‘many’) has enabled us to draw attention to regularities/irregularities in the data. As stated in our limitations, this approach is not meant to convey generalizability beyond the study population
It’s unclear to me why some of the direct quotes are integrated into the paragraphs and some are separated out. It does not seem to be related to the length of the quote.	Quotes have been integrated into the text to help with flow and structure of findings. At times, shorter quotes have been integrated into paragraphs to maximise integration of participant data without consuming word count.
I appreciate the narrative that the authors crafted in describing the qualitative results – each section leads well into the next.	Thank you!

On p. 16, line 52, the authors write “whilst deemed necessary” about the measures of participants separating from loved ones. The authors should clarify that the participants deemed this necessary, if that was the case.	Thank you for this clarification. We have amended the text to: “Such measures, whilst deemed necessary by participants, induced additional psychosocial strains including loneliness and isolation”:
Discussion	
The authors did a wonderful job writing this discussion overall.	Thank you! 😊
On p. 22, line 27, the following claim needs a citation: “Consistent with research with HCWs, feeling unsafe and vulnerable to infection are predictive of poor mental health.”	Thank you for pointing this out. We have added in references here: “Consistent with research with health and care workers, feeling unsafe and vulnerable to infection are predictive of poor mental health”⁹²⁶ “
On p. 23, line 15, I just wanted to clarify whether the authors meant “provision of protective measures by employees” or “by employers.” It seems that “employers” makes more sense if institutional provision of these measures would decrease stress and isolation. But perhaps the authors mean that when the employers are not providing this and the employees themselves provide them (e.g. hand sanitizer, etc.), it has the same outcome on mental health.	Thank you for highlighting this. We do in fact mean ‘employers’, and this has been amended: “The provision of protective measures by employers”
On p. 23, line 28, I appreciate that the authors brought up that a theme of workplace unity was not found in this study. It would be helpful to expound on this a little as to the reasons why that might be. For instance, it seems like many of the jobs represented in the study were more solitary positions (e.g., bus drivers) that may not engender as much of a sense of community.	Thank you for highlighting this important point. We have restructured this paragraph to make it clear that ‘increased workloads...elevated feelings of stress and subsequent workplace tension and conflict’ were detrimental to workplace unity. We have also added in a further finding around a lack of internal recognition for work, that may also have contributed here: “Workplace challenges also posed several additional stressors. Increased workloads were common and led to elevated feelings of stress and subsequent workplace tension and conflict. Some participants also reported limited internal recognition for their work and felt that the risks they were exposed to were not fully acknowledged by senior staff”
On p. 24, line 23, a stronger statement could be made – the timing of the study should be considered when interpreting the findings, particularly because of the rapidly evolving/changing nature of COVID-19 over the past year. The authors note at this point in the discussion that the majority of interviews were completed between September and November 2020. It would be helpful to add to Table 1 a frequency of which month the interviews took place in from September 2020 to January 2021 to have a better sense of the context of the	Thank you for highlighting this. We agree the context in which the interviews were conducted needs further clarity. As such, we have added to Table 1 the frequency of which month the interviews took place. Please also note a minor amendment to the first paragraph of methods. This is meant to say ‘the majority of participants were interviewed between September 2020 – Jan 2021’. We have therefore amended to say ‘interviews were conducted between July 2020 – January 2021’

majority of these interviews.	
On p. 24, line 34, the participants were motivated “and” willing (rather than “or”), given that they did in fact participate. On line 39, data is plural, so the verb should be “cover” instead of “covers.”	Thank you for highlighting this. Both suggestions have been amended in the text. “Second, this study may be limited by a sample biased toward those motivated and willing to participate. There is the potential that the views and experiences of those unable or unwilling to participate may differ from those in this study (e.g. unaffected by working conditions) and have therefore not been documented. Finally, our data cover a range of keyworker occupations”
The authors make the comment on p. 24, line 43 that “Where possible, we have attempted to draw out any distinctions between occupations in the data.” I am not sure what they are referring to exactly, and it would be helpful to discuss any distinctions (whether in the Results or Discussion), particularly because a large percentage of the sample was bus drivers, compared to other occupations that generally had only 1 participant.	To clarify, we have drawn out distinctions between different occupations in the findings, where they have occurred. For example, page 13 (‘In some workplaces, such as on buses and in supermarkets, other protective measures including daily antiviral cleaning and enhanced sanitation were often inadequate’) and page 17 (‘those who transitioned to online working (including police, teachers and bank workers welcomed such changes, but noted difficulties’) We have also amended in discussion (p.23) to say: “Where possible, we have attempted to draw out any distinctions between occupations in the reporting of our results”
On p. 25, line 42, the authors state that “it is worth noting that many of the psychological demands experienced by keyworkers and highlighted in this article existed well before the advent of COVID-19, including occupational stress ³⁴⁻³⁶ , low levels of job satisfaction ^{35 37} and burnout ³⁸ .” While this is an interesting idea to bring up and could be relevant for future implications, I am not sure whether this is supported by the findings. It would be helpful to add an example or two here as to how the data support this conclusion.	Thank you for this suggestion. We have since revised this sentence to make it clear that some of these psychological demands have been reported in previous studies conducted before the pandemic and are similar, rather than the same, as the findings in our study: “Finally, while these measures may help mitigate the immediate psychological effects of the pandemic, it is worth noting that previous research conducted before the pandemic has identified similar psychological demands among keyworkers to those highlighted in this article , including occupational stress^{39-41”}
Reviewer 2	
Observations: 1. The topic chosen for the investigation is utmost significant in the present day scenario. The design of the investigation is indeed one of the pioneer efforts. 2. The rationale of the study is well-written. 3. The procedure for sample recruitment was done appropriately. Ethical criteria were addressed. 4. The process of theoretical saturation was accurately followed for data collection. 5. Empirical process was followed for data analysis. All the themes and sub-themes	Thank you for your observation and comments. We appreciate your kind feedback and have made amendments to the paper based on your suggestions, which have substantially improved the overall quality of the paper.

have been analyzed meticulously. 6. The strengths and limitations of the study have been explained with clarity. 7. All the sub-themes have been adequately addressed in the discussion. 8. References are appropriate and well researched.	
In the abstract on page no. 03 line no. 28 five professions have been mentioned whereas in the findings on page no. 11 line nos. 17-34 thirteen professions have been mentioned. Therefore in the abstract either the terms 'and others' may be incorporated with the previously mentioned five professions or line no. 25 on page no. 03 may be rephrased and written as '... in a range of 13 non-healthcare keyworker occupations...'	Thank you for the suggestion. We have now added in all professions and sectors to the abstract to clarify: 23 participants aged 26-61 (mean age =47.2) employed in a range of non-healthcare keyworker occupations, including transport, retail, education, postal services, the police and fire services, waste collection, finance and religious staff
Age- range of the sample should be mentioned in the abstract.	Thank you for the suggestion. We have added age range to abstract.
On Page no. 15 line nos. 50-55 and Page no.16 line nos. 03-11 the investigators have very well addressed the 'feelings of loneliness and isolation of the participants' very well. However, one or two interviews of the participants depicting that they are missing their children may enrich this finding further.	Thank you for the comment. Although we appreciate this could have been an issue, we do not have any data of participants specifically stating that they miss their children – instead described missing 'family' or 'loved ones'.
Few more references (03-05) may be incorporated in the discussion.	Thank you for the comment We have added in 3 more references throughout the discussion, and we hope that this now makes the discussion richer. Paul E, Mak HW, Fancourt D, et al. Comparing mental health trajectories of four different types of key workers with non-key workers: A 12-month follow-up observational study of 21,874 adults in England during the COVID-19 pandemic. medRxiv 2021:2021.04.20.21255817. doi: 10.1101/2021.04.20.21255817 Chatterji S, McDougal L, Johns N, et al. COVID-19-Related Financial Hardship, Job Loss, and Mental Health Symptoms: Findings from a Cross-Sectional Study in a Rural Agrarian Community in India. Int J Environ Res Public Health 2021;18(16) doi: 10.3390/ijerph18168647 [published Online First: 2021/08/28] 37. Posel D, Oyenubi A, Kollamparambil U. Job loss and mental health during the COVID-19 lockdown: Evidence from South Africa. PLOS ONE 2021;16(3):e0249352. doi: 10.1371/journal.pone.0249352
Few more studies from Asian, African and South American countries may be referred (if possible) for the present	Thank you for this suggesting this important point. Following a search for further studies, we located two studies that can be incorporated into our study – one

empirical study and incorporated in the references.	from India and another from South Africa: 36. Chatterji S, McDougal L, Johns N, et al. COVID-19-Related Financial Hardship, Job Loss, and Mental Health Symptoms: Findings from a Cross-Sectional Study in a Rural Agrarian Community in India. Int J Environ Res Public Health 2021;18(16) doi: 10.3390/ijerph18168647 [published Online First: 2021/08/28] 37. Posel D, Oyenubi A, Kollamparambil U. Job loss and mental health during the COVID-19 lockdown: Evidence from South Africa. PLOS ONE 2021;16(3):e0249352. doi: 10.1371/journal.pone.0249352 We have included studies from Asia throughout our paper, e.g.: 8. Liu Q, Luo D, Haase JE, et al. The experiences of health-care providers during the COVID-19 crisis in China: a qualitative study. The Lancet Global Health 2020;8(6):e790-e98. doi: 10.1016/S2214-109X(20)30204-7 9. Lai J, Ma S, Wang Y, et al. Factors Associated With Mental Health Outcomes Among Health Care Workers Exposed to Coronavirus Disease 2019. JAMA Network Open 2020;3(3):e203976-e76. doi: 10.1001/jamanetworkopen.2020.3976 10. Sun N, Wei L, Shi S, et al. A qualitative study on the psychological experience of caregivers of COVID-19 patients. Am J Infect Control 2020;48(6):592-98. doi: 10.1016/j.ajic.2020.03.018 [published Online First: 2020/04/08] 11. Kang L, Li Y, Hu S, et al. The mental health of medical workers in Wuhan, China dealing with the 2019 novel coronavirus. Lancet Psychiatry 2020;7(3):e14. doi: 10.1016/s2215-0366(20)30047-x [published Online First: 2020/02/09]
Reviewer 3	
	Thank you for reviewing our paper, as well as your kind comments regarding the value of the work. We appreciate your feedback and suggestions, which we have addressed below. This has helped improve the overall quality of the paper.
It would have been interesting to have included your semi structured interview guide so that readers can see what the specific focus was during your interviews.	Thank you for the suggestion. Further information regarding the interview topic guide development is now included in the methodology, and some specific questions are now included in Figure 2. For the full topic guide, this is included as supplementary material.
One tiny point is that , at times, you refer to data in the singular - as I say a tiny point to consider.	Thank you for highlighting. We have amended referencing to data in the singular to plural (e.g p.23. Finally, our data cover a range of keyworker occupations)

1. Vera San Juan N, Aceituno D, Djellouli N, et al. Mental health and well-being of healthcare workers during the COVID-19 pandemic in the UK: contrasting guidelines with experiences in practice. *BJPsych Open* 2021;7(1):e15. doi: 10.1192/bjo.2020.148 [published Online First: 2020/12/10]
2. Vindrola-Padros C, Andrews L, Dowrick A, et al. Perceptions and experiences of healthcare workers during the COVID-19 pandemic in the UK. *BMJ Open* 2020;10(11):e040503. doi: 10.1136/bmjopen-2020-040503
3. Sun N, Wei L, Shi S, et al. A qualitative study on the psychological experience of caregivers of COVID-19 patients. *Am J Infect Control* 2020;48(6):592-98. doi: 10.1016/j.ajic.2020.03.018 [published Online First: 2020/04/08]
4. Liu Q, Luo D, Haase JE, et al. The experiences of health-care providers during the COVID-19 crisis in China: a qualitative study. *The Lancet Global Health* 2020;8(6):e790-e98. doi: 10.1016/S2214-109X(20)30204-7
5. Lai J, Ma S, Wang Y, et al. Factors Associated With Mental Health Outcomes Among Health Care Workers Exposed to Coronavirus Disease 2019. *JAMA Network Open* 2020;3(3):e203976-e76. doi: 10.1001/jamanetworkopen.2020.3976
6. Kang L, Li Y, Hu S, et al. The mental health of medical workers in Wuhan, China dealing with the 2019 novel coronavirus. *Lancet Psychiatry* 2020;7(3):e14. doi: 10.1016/s2215-0366(20)30047-x [published Online First: 2020/02/09]
7. Aughterson H, McKinlay AR, Fancourt D, et al. Psychosocial impact on frontline health and social care professionals in the UK during the COVID-19 pandemic: a qualitative interview study. *BMJ Open* 2021;11(2):e047353. doi: 10.1136/bmjopen-2020-047353
8. Lan F-Y, Suharlim C, Kales SN, et al. Association between SARS-CoV-2 infection, exposure risk and mental health among a cohort of essential retail workers in the USA. *Occupational and Environmental Medicine* 2020:oemed-2020-106774. doi: 10.1136/oemed-2020-106774
9. Cai M, Velu J, Tindal S, et al. 'It's Like a War Zone': Jay's Liminal Experience of Normal and Extreme Work in a UK Supermarket during the COVID-19 Pandemic. *Work, Employment and Society* 2020:0950017020966527. doi: 10.1177/0950017020966527
10. Paul E, Mak HW, Fancourt D, et al. Comparing mental health trajectories of four different types of key workers with non-key workers: A 12-month follow-up observational study of 21,874 adults in England during the COVID-19 pandemic. *medRxiv* 2021:2021.04.20.21255817. doi: 10.1101/2021.04.20.21255817
11. The Lancet. The plight of essential workers during the COVID-19 pandemic. *The Lancet* 2020;395(10237):1587. doi: 10.1016/S0140-6736(20)31200-9
12. Goldblatt P, Morrison J. Initial assessment of London bus driver mortality from COVID-19. London: UCL Institute of Health Equity, 2020.

VERSION 2 – REVIEW

REVIEWER	Reilly, Shannon E University of Virginia Health System, Neurology
REVIEW RETURNED	05-Oct-2021
GENERAL COMMENTS	Thank you for the opportunity to review the revision of this manuscript, 'Stressed, uncomfortable, vulnerable, neglected': a qualitative study of the psychological and social impact of the COVID-19 pandemic on UK frontline keyworkers." The authors did a wonderful job of responding to reviewer comments. I commend them, and I recommend acceptance for publication. One minor remaining comment is that I recommend that in Table 1, the authors change "ethnicity" to "self-identified ethnicity." Thank you again for the opportunity to review this timely, important work!